# A lightweight and robust authentication scheme for the healthcare system using public cloud server

Irshad Ahmed Abbasi[1,2]*, Saeed Ullah Jan[3], Abdulrahman Saad Alqahtani[4], Adnan Shahid Khan[2], Fahad Algarni[4]

**1** Department of Computer Science, College of Science and Arts Belqarn, University of Bisha, Sabtul-Alaya, Saudi Arabia, **2** Faculty of Computer Science and Information Technology, Universiti Malaysia Sarawak, Kota Samarahan, Malaysia, **3** Higher Education Department of Khyber Pakhtunkhwa at Govt. College Wari Dir Upper, Wari, Khyber Pakhtunkhwa, Pakistan, **4** Department of Computer Science, College of Computing and Information Technology, University of Bisha, Bisha, Saudi Arabia

☯ These authors contributed equally to this work.
* aabasy@ub.edu.sa

**Data Availability Statement:** All relevant data are within the paper and its Supporting Information files.

## Abstract

Cloud computing is vital in various applications, such as healthcare, transportation, governance, and mobile computing. When using a public cloud server, it is mandatory to be secured from all known threats because a minor attacker's disturbance severely threatens the whole system. A public cloud server is posed with numerous threats; an adversary can easily enter the server to access sensitive information, especially for the healthcare industry, which offers services to patients, researchers, labs, and hospitals in a flexible way with minimal operational costs. It is challenging to make it a reliable system and ensure the privacy and security of a cloud-enabled healthcare system. In this regard, numerous security mechanisms have been proposed in past decades. These protocols either suffer from replay attacks, are completed in three to four round trips or have maximum computation, which means the security doesn't balance with performance. Thus, this work uses a fuzzy extractor method to propose a robust security method for a cloud-enabled healthcare system based on Elliptic Curve Cryptography (ECC). The proposed scheme's security analysis has been examined formally with BAN logic, ROM and ProVerif and informally using pragmatic illustration and different attacks' discussions. The proposed security mechanism is analyzed in terms of communication and computation costs. Upon comparing the proposed protocol with prior work, it has been demonstrated that our scheme is 33.91% better in communication costs and 35.39% superior to its competitors in computation costs.

## 1. Introduction

The effective handling of stored information gathered from different patients has widely been implemented for research, investigation, and treatment in the healthcare system. This sensitive data is collected with the help of wearable devices embedded inside the human body. The network-enabled devices are connected to the public network over several methods like IEEE

**Funding:** The authors are thankful to the Deanship of Scientific Research, University of Bisha (SA) for supporting this work through grant number UB-GRP-65-1444.

**Competing interests:** The authors have declared that no competing interests exist.

802.15.4 port, IEEE 802.16, WiFi, or WiMAX [1]. The sensor has limited low-power battery storage capacity while performing high computation by generating tremendous output, which requires substantial computing power, massive storage capacity, and real-time processing [2]. For this purpose, a public cloud server offers affordable, flexible, high-performance computing, virtualized storage, and software applications for the healthcare system or patient at home [3]. It is cost-effective, scalable, and available for data-driven pervasive healthcare systems; other service providers can also take benefit by demanding the same high-speed online services from it so that to provide high-quality treatment, effective communication of healthcare personnel with patients, doctors, nurses, pharmacists, and other staff members [4].

A medical information system stored in public cloud services can support the healthcare system for numerous delivery services. This facility is made possible by physiological monitoring devices for patients at home directly or with the doctors at a clinic or in the e-healthcare industry [5]. The public cloud server is mature for the interaction and enhanced sharing of valuable information between various medical institutions, hospital systems, and respective care providers. Such a healthcare system is preferred to reduce costs and make efficient processes, preserving the medical record's privacy and patient's identity [6]. Patients' concerns about losing their privacy are a significant hurdle to adopting cloud-based healthcare systems since they may feel uneasy and lack confidence in the service providers to keep their identities a secret. The transmission of patient information through an insecure internet needs to be secure, and patient privacy is preserved [7]. Thus, a public cloud server's stored medical information system needs a strong authentication technique to protect accessibility, confidence, and authenticity [8].

Similarly, cloud computing must be implemented to meet the tremendous output generated by numerous IoT in the healthcare system, which requires constant availability and storage [9]. Cloud computing is a potential paradigm in computing that transfers the hardware and software platform to outside service providers (e-healthcare systems) who provide the healthcare facilities to its users (patients) at a significantly lower cost. Cloud computing has much potential for improving the e-healthcare system to ease end-users lives—the cloud transfers patient-sensitive data to healthcare enterprises for management by physicians, labs, research and associated tasks. In general, an e-health cloud is a platform that manages and stores vast amounts of health data from various healthcare providers [10].

Furthermore, attention is required owing to the ubiquitous output of thousands of wearable devices in the healthcare system and the production dispatch for storing it in the public cloud server. As thousands of Internet-of-Medical-Things (IoMT)/sensors generate the result, transmitted to the server for storage requires proper authentication; otherwise, no one builds trust in it due to a lack of privacy, security, and continuous changes in patient conditions which may cause massive mobility issues [11]. All these issues and challenges are due to the need for an appropriate authentication scheme. Therefore, a reliable authentication protocol for such a sensitive environment is much needed to support the accurate authentication of each device, exact data stored in the server, low end-to-end delay, integrity, confidentiality, and low energy consumption. So that, to ensure the privacy of patient security of stored information and deny control of the resources by any fraudulent user. The main contributions of this research work for such a system are as follows:

- An ECC-based lightweight authentication protocol has been proposed to securely provide services to the end-user in the cloud-enabled healthcare system.

- A fuzzy extractor method is used to design the proposed protocol to make the authentication process more secure, and the system shows uniqueness while performing any task. An adversary cannot forge, extract, or collide the hash image generated from user biometrics in combination with a random key extracted before authentication.

- The BAN logic and ROM security proofs were carried out, demonstrating that the proposed protocol is verifiably secure.

- The key secrecy, integrity, confidentiality, and reachability have been verified through a well-known software verification toolkit, ProVerif.

## 2. Literature survey

The growing need and other advancements in computing technology can provide scalable services with the potential to up-size or downsize information storage for cloud computing, which is frequently utilized in the healthcare system. In this regard, Zhang et al. [6] proposed a disease prediction scheme using a cloud server for the healthcare system in which they claimed that security and privacy are two major concerns for such a system. Their strategy was based on the neural network using a single-layer perception (SLP) algorithm containing double layers (input and output). However, their scheme doesn't fulfil all the security functionalities for participants. Bhatia and Malhotra [7] proposed a scheme based on Morton filters using cloud computing. They claimed that most of the security schemes for patient diagnosis are designed on cuckoo or bloom filters, which suffer from security and privacy issues. However, using their scheme, patients cannot preserve his/her privacy. Sivan and Zukarnain [8] proposed a cloud-based healthcare system using AI (Artificial Intelligence) and ML (Machine Learning) techniques. They addressed privacy and security-related issues and challenges in their research for secure access of patient sensitive information by physicians, labs and hospitals. They also suggested that approximate security solutions can beneficially mitigate all the issues mentioned earlier in the cloud-based healthcare system. Still, they failed to test their methodology in a real-world environment. Chenthara et al. [9] proposed a collaborative mechanism for the healthcare system that can support MPPDS (multi-level privacy-preserving data sharing). However, in some layers, the strong adversary can easily capture data and later be used for replay and DoD attacks.

Huang et al. [10] proposed a security mechanism for MHSN (mobile healthcare social network) based on identity using cloud computing. In their scheme, a user via a mobile device can browse their health information from the cloud and share it with medical professionals for possible diagnosis. However, such a practice cannot be feasible in different countries; their scheme is weaker against MIMT attacks. Li et al. [11] designed a searchable private key cryptographic-based authentication scheme for medical data using cloud computing. Their strategy has dynamically searched a patient's medical profile in encrypted form over the symmetric key and mitigated the key-sharing drawback. But due to this, the computation costs go high, which in turn doesn't get the patient's attention due to the slow computation of patient-sensitive data in the form of X-ray images, ECG, and EEG. Ma et al. [12] proposed a multi-access attribute-based encryption (MA-ABE) scheme for people who desire to restrict their continuous hospital visits for diagnosis. It still doesn't provide efficient and effective services to the community due to low performance. Nguyen et al. [13] proposed a blockchain-based secure authentication scheme for electronic healthcare records (EHRs). They claimed that their strategy provides low operational costs, availability, and flexibility. However, using cryptographic primitives for blockchain technology can degrade the performance metrics for end users/patients. Chen et al. [14] demonstrated that to preserve patient-sensitive information (privacy), electronic medical records must be secure from intruders. In this regard, they proposed a framework that offered authorization, confidentiality, and integrity of records efficiently. However, on one side, they secure the record, while on the other, their performance degrades. Wu et al. [15] used a simple cryptographic hash function to design a scheme to secure patient information through cloud

**Table 1. Critical literature review.**

| Ref# | Year | Approach | Advantages | Limitations |
|---|---|---|---|---|
| [16] | 2016 | Healthcare devices can be connected to the cloud server for numerous tasks. | Excellent performance achieved | However, when the connection is dropped, the outputs generated by the devices are a soft target for attackers. |
| [17] | 2016 | A symmetric cryptographic algorithm and crypto hash function-based authentication scheme for IoT-based e-healthcare system. | Third-party involvement makes the system credible for integrity, confidentially and availability of physiological parameters. | Rigorously, their scheme can be feasible for two to three parties/IoT; when the number of sensor nodes increases, their scheme doesn't perform well, and authentication takes maximum computation costs. |
| [18] | 2017 | Cryptographic Algorithms | Mitigated ambulance incidents, heart attacks, and brain stroke. | The response time is much slower for such sensitive data broadcasting. |
| [19] | 2017 | A security mechanism that secures the transmission between fog and healthcare devices. | Effectively alleviate the interoperability challenges between cloud and edge computing paradigms. | However, due to the non-usage of the cryptographic approach, their strategy can easily be hacked. |
| [20] | 2018 | ECC | It efficiently mitigated the stolen verifier, impersonation, and insider attacks. | The user is traceable. |
| [21] | 2019 | Blockchain | Successfully achieved authentication | Vulnerable to insider and sniffing attacks |
| [22] | 2019 | Zero-knowledge concept and fuzzy extractor | Successfully segmenting the patient parenchyma of CT lung images | The security analysis section is missing |
| [23] | 2019 | certificate-less cryptography | Achieved anonymity and privacy | Feasible for only two-party communications. |
| [24] | 2019 | fuzzy extractor | No one is cracking/guessing the biometrics of a user. | Vulnerable to password-guessing attacks. |
| [25] | 2020 | MBFTA (mixed Byzantine fault tolerance algorithm) | Efficient transaction handling, anonymity, and privacy are achieved. | Data obfuscation and malicious operations might have yet to be tackled. |
| [26] | 2020 | Message Authentication Code (MAC) Secure Hash Algorithm (SHA-1) | The researchers have seriously used a one-time biometric key +MAC-SHA1 along with random mapping – which means the adversary doesn't violate the security features while using their scheme. | However, it is feasible for one-party authentication when the number of nodes/entities increases; their scheme, which is based on MAC-SHA1, cannot be feasible. |
| [27] | 2022 | Batch authentication algorithm | Balance of security with performance has been seen. | Batch verification of messages is a little bit heavyweight |
| [28] | 2023 | A biometric-based key validation | The researchers have efficiently utilized the low latency WBAN for deployed cryptographic primitives. | However, prompt data delivery causes disastrous results, potential threats to patient freedom and data consistency issues. |
| [29] | 2023 | ECC for IoT-enabled cloud server | The researchers have used KeyGen(.), SigGen(.), GenProof(.) and VerifyProof(.) algorithms, which the adversary cannot break the session key during its establishment for secure communication. | However, due to using a smart device to store security credentials, adversaries can easily launch a stolen verifier attack on their scheme. |

computing. But a simple hash cryptographic function can provide secure services to a single user; when the number of users/patients increases, their scheme isn't feasible; therefore, it fails to provide efficient and reliable services. The remaining related literature review in the form of a table is also shown in Table 1.

Therefore, considering the literature survey, it has been observed over the recent years, that different researchers have proposed numerous schemes at different times. These schemes are either based on bilinear pairing, discrete logarithm problems, RSA, or other cryptographic techniques with high communication/computation costs due to exponential execution complexity or suffer from security and privacy issues, most of which are unable to withstand the known vulnerabilities. In this regard, we have proposed a scheme in section 4 of the research paper based on ECC, using fuzzy extractor method that can offer better security and efficient performance then recent schemes.

## 3. Preliminaries

This section will demonstrate the basic concept for designing the proposed authentication schemes. These foundation terminologies are described as under:

## 3.1 System architecture

The architecture aims to secure the cloud-based patient remote monitoring without losing privacy and performance. So, the architecture consisted of tiny sensors affixed to the patient's body to collect physiological parameters. These network-enabled sensors are connected to an external network or other devices through IEEE 802.15.4 port and then to the main network to enable external connectivity. So far, two participants are involved in the proposed system architecture are public cloud server and patient/end-user. These are described as under:

*Public Cloud Server (PCS)*: A system containing unlimited storage capacity for storing hounds of hundreds of medical data generated by millions of wearable devices embedded inside the human body. Hospitals that desire to operationalize their healthcare system into a cloud-based system and connect numerous wearable devices can outsource physiological data from a public cloud server. The public cloud server can also have strong computation capabilities for the stored sensitive patient information, learning, and diagnosis prediction.

*Patient or User (U)*: A person who accesses the entire system takes medical data from either doctors or embedded sensors and then sent onward to the hospitals for treatment. All patients must trust the healthcare system as it provides identity to the patient and can also be responsible for the diagnosis, treatment, and real-time monitoring. Hospitals must also offer intelligent services to patients to securely check their status and online suggested treatment, disease confirmation, and different symptoms, as shown in Fig 1.

## 3.2 Elliptic Curve Cryptography (ECC)

This type of asymmetric key cryptography or public key cryptography is based on a curve over a finite field of equation defined in the form $y^2=x^3+ax+b$ over an $F_q(a, b)$ (finite field). ECC [30] has the following unique features:

- It is also called NP-problem, which is hard for adversary $A$ to break, whereas a, b set of finite field elements on the curve. Let a point in the curve is P, and its base is Q; then *Q=s.P* which is computationally difficult to find *s* (integer value), means infeasible.

- If we draw a chord that interests the curve at a third point, the result reflects on the x-axis and is represented by –R.

- It offers more efficient security than RSA.

- By giving points xP, yP, P over $F_q(x, y)$, it is much more challenging to calculate xyP.

- Let $(a, b)\in E$ and $a\neq b$, then $\Rightarrow R=a+b$. So according to the curve equation in ECC, $x^3=\lambda^2$-$x$-$x^2$, $y^3=\lambda(x+x^3)$-$y$ *whereas* $\lambda=(y^2$-$y)/x^2$-$x)$.

- Let a point P is doubled in the curve such that $P\in E$ whereas $P\neq$-$P$, then $\Rightarrow R=2P$. So according to ECC, $x^3= \lambda^2$-$2x$, $y^3= \lambda(x+x^3)$ *whereas* $\lambda=(3x^2$-$a)/2y$.

- ECC is an interesting asymmetric approach that offers greater security with a smaller key size than RSA. For example, if the key length in RSA becomes 1024, then the same key will be 160 bits in ECC. Similarly, if the ECC key size is 256 bits, it will be 3072 bits in RSA. The ratio of ECC and RSA is 1:6, meaning ECC is six times smaller than RSA and offers excellent security.

## 3.3 Fuzzy extractor

Universally, academia and industries are using the conventional cryptographically generated key for authentication, which could be a better practice from a security point of view. By considering the more vigorous adversary nature, biometrics in combination with conventional

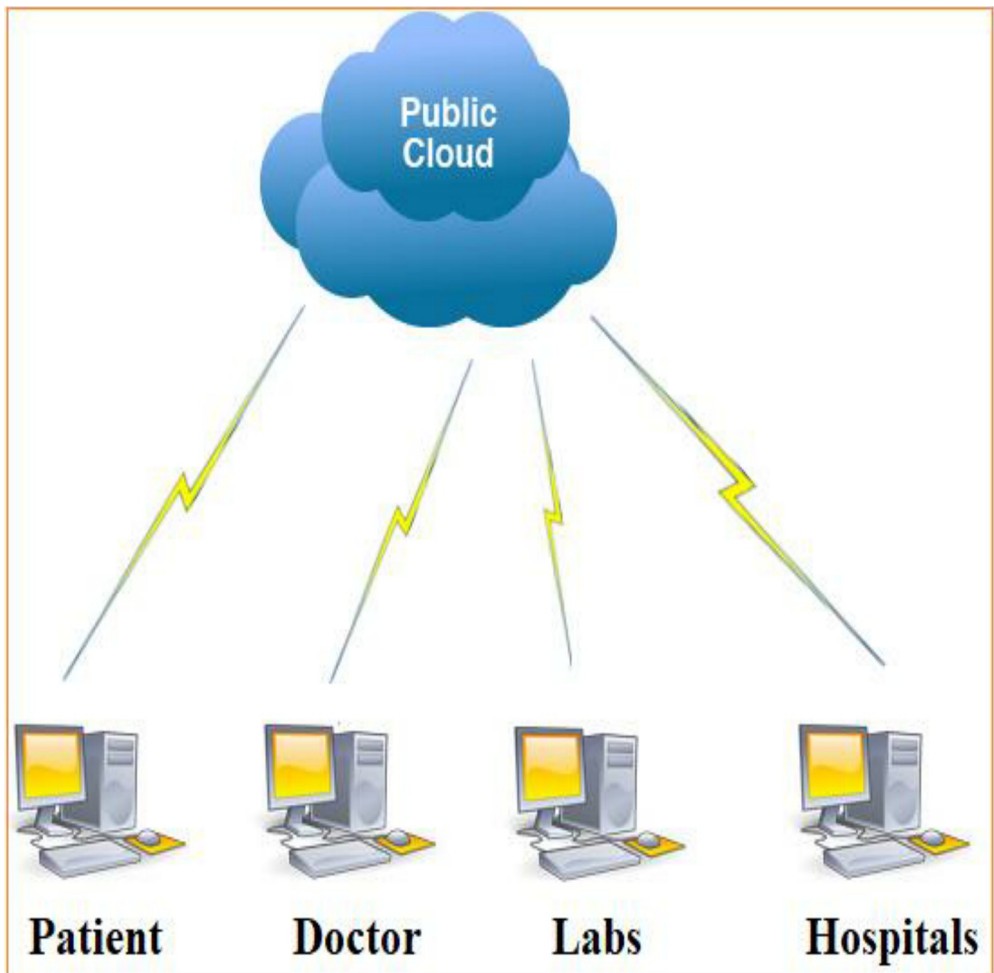

**Fig 1. System architecture.**

keys are used but using constant iris or fingerprint, which could be a better practice. An adversary can easily forge or duplicate during authentication to collide with the original image/key. Therefore, a fuzzy extractor method is used to make the authentication process more secure, and the system shows uniqueness while performing any task [31].

### 3.4 Threat and adversary model

If adversary $A$ obtains the secret session key of a legitimate user/patient and desires to act as a malicious user in spoofing the actual user, adversary $A$ plays the role of both active and passive attacker by detecting the exchanged information among the user and server and eavesdropping fake information [32] in the following manner:

1. $A$ may disclose/target the session key's integrity by changing its parameters.

2. $A$ might modify the message exchanged publically among the user, server, and vice-versa.

3. $A$ might disrupt valuable services among legitimate peers.

4. $A$ might gain control of the wireless line/public line for accessing exchanged messages and obtaining its internal secret credential.

5. $A$ can also have the power to malfunction the public channel among participants.

6. $A$ guess is either the password or ECC key is threatening the confidentiality of the message.

7. $A$ can launch a brute force attack, cryptanalyze a scheme, or use a guessing attack to find the right secret login credentials.

8. $A$ might discover the secret credential through cryptanalysis technique.

9. $A$ can have the power to stop a legitimate user or public cloud server for the established session.

10. $A$ exhausts the computation, storage, and other resources to stop working in the communication process.

## 3.5 Review analysis of Azrour et al. scheme

In 2021 Azrour et al. [33] proposed a scheme for remote healthcare authentication through cloud-enabled IoT. They [33] presented their strategy in 5 phases: step, sensor registration, user registration, login & authentication phase, and password update phase. These phases are described one by one as under:

1. **Setup Phase:** In this phase, the chief executive of a healthcare system selects a key for public cloud server $X_s$, hash $h(.)$, and the server publishes the hash code and keeps it in its memory for future correspondence.

2. **Sensor Registration Phase**: The server chooses identity for the sensor $Id_{sn}$, unique key $K_{CS-SNi}$, computed $SK=h(Id_{sn}||K_{CS-SNi})$, saved $Id_{sn}$ and $HSK=SK\oplus h(X_s||Id_{sn})$ in its memory.

3. **User Registration Phase:** The user first selected their identity $Id_i$, picked two random numbers, password $pw_i$ computed $MID=h(Id_i||a)$, $MPW=h(Id_i||b)$ and transmitted {MID, MPW} to server over a secure channel. The server SC picked a random number c, computed $V=h(MID||X_s)\oplus h(MPW||c)$, saved MID, c, and sent V toward the user. The user stored {V, a, b, MID} in its smart card.

4. **Login & Authentication Phase:** The user provided their smart card into a terminal and entered $Id_i$, $pw_i$, the smart card confirming $MID=h(Id_i||a)$ with the already stored smart card values, if found valid, picked A, computed $x=V\oplus h(h(Id_i||pw_i||b)||c)$, $V_1=h(x||A)$ and sent {$V_1$, MID, A, $Id_{sn}$, $T_1$) towards cloud server. The cloud server checked timestamp $T_2-T_1\leq\Delta T$, computed $w_1=h(MID||X_{s=)}$, verified $V_1=h(w_1||A)$, picked B, calculated $w_2=HSK\oplus h(Id_{sn}||X_s)$, $HID=h(MID||Id_{sn})$, $V_2=h(HID||w_2||T_2||B)$ and sent {$V_2$, B, MID, $T_2$} towards sensor over open channel. The checked time stamp $T_3-T_2\leq\Delta T$, calculated $HID^/=h(MID||Id_{sn})$, checked $V_2=h(HID^/||SK||T_2||B)$, if found validated, SN picked C, computed $V_3=h(MID||Id_{sn}||SK||T_3||C)$ and sent {$V_3$, C, HID, $Id_{sn}$, $T_3$} back towards cloud server over an insecure channel. The SN checked timestamp $T_4-T_3\leq\Delta T$, confirming $V_3=h(MID||w_3||T_3||C)$, picked D, calculated $V_4=h(w_1||MID||Id_{sn}||T_4||D)$, $S_{key}=h(w_1||MID||Id_{sn})$ and sent {$V_4$, D, $Id_{sn}$, $T_4$} back towards user. The user, too, checked timestamp $T_5-T_4\leq\Delta T$ and confirmed $V_4=h(x||MID||Id_{sn}||T_4||D)$; if validated, the user too computed $S_{key}=h(x||MID||Id_{sn})$ and kept as secret session key.

5. **Password Update Phase**: The user typed $Id_i$, $pw_i$, validated $MID=h(Id_i||a)$ and will be asked to enter a new password $pw_i^*$, by choosing two numbers $a^*$, $b^*$ and computed $MID^*=h(Id_i||a^*)$, $MPW^*=h(Id_i||pw_i^*||b^*)$ and encrypted the message $M_u=E_{SK}(MPW||MPW^*||$

MID||MID*||V) and sent towards the cloud server. The cloud server decrypted using the same key SK $M_u^{/}$=$D_{SK}$(MPW||MPW*||MID||MID*||V), checked V=h(MID||$X_s$)⊕h(MPW|| c), if confirmed, the cloud server picked c*, replaced MID, c with MID*, c*, computed V*=h(MID*||$X_s$)⊕h(MPW*||c*), $M_s$=$E_{SK}$(V*) and sent back to user. The user decrypted $M_s^{/}$=$D_{SK}$(V*) and replaced V, a, b, MID with V*, a*, b*, MID*.

## 3.6 Weaknesses in Azrour et al. scheme

After the critical review analysis of the Azrour et al. [33] scheme, the following loopholes/ weaknesses have been noticed:

1. *Insider Attack*: Due to not using a biometric/fuzzy extractor, an attacker can efficiently compute MID=h($Id_i$||a), x=V⊕h(h($Id_i$||$pw_i$||b)||c), and $V_1$=h(x||A) by taking any two random numbers. After doing such computations, the attacker enters internally into the server to launch an insider attack.

2. *DoS Attack*: The $Id_{sn}$ is transmitted openly through a public network channel; an attacker can easily intercept the line and copy it for malicious deeds.

3. *Replay Attack*: If an attacker from the open channel captures the identity because it is transmitted openly, they may use it for a potential replay attack at some other time.

4. *Privileged-insider Attack*: In this scheme, in each round trip, a lot of random numbers, like a, b, c, c*, A, B, C, D, and $X_s$ have been taken, which a privileged user can use for identifying the real identity of the user, server or sensor and launch personal insider attack for other credential hacking. More explicitly, a privileged user sitting on the system can easily use these numbers to hack the whole system at some other time for other websites or malicious deeds. Here, the researchers have observed that if the system generates numerous random numbers for each session key establishment each time, there is a chance that the system can launch an attack on its own credentials.

5. Finally, in Azrour et al. [33] scheme, the researchers needed to provide the facility for revocation/re-registration.

## 4. Proposed protocol

This section of the article demonstrates the proposed protocol, which consists of the following phases. In contrast, the various notations and descriptions used for designing the protocol are shown in Table 2.

**Table 2. Notations and their description.**

| Notation | Description | Notation | Description |
|---|---|---|---|
| U | User | PCS | Public Cloud Server |
| $E(F_q)$ | Curve | s | Server secret number |
| q | Prime Order | O | Base Point of Curve |
| $ID_P$ | User Identity | $PW_P$ | User Password |
| $B_P$ | User Biometrics | G(.) | Biometric Generation Function |
| Rep(.) | Biometric Replication Function | r, $r_P$, $r_{PCS}$ | Random numbers |
| a,b | Curve Points | || | Concatenation Function |
| $E_s$(.) | Encryption on s | $Dec_s$(.) | Description on s |
| ⊕ | XOR Operation | T | Timestamp |

## 4.1 Setup phase

Initially, the system creates different parameters for its corresponding participants and then sent according to the requirement. A public cloud server ($S_{PCS}$) first selects a curve $E(F_p)$ at point p of the base point Os of order prime q. Second, the $S_{PCS}$ chooses hash function h(.), secret number $s \in Z^*_q$, and computes $P_{key}=s.O$ as the public key, stores $s$ securely in its memory, and publicize $\{O, q, E(F_p). h(.)\}$ parameters.

## 4.2 Registration phase

In this phase, anyone who desires to access SPCS (Public Cloud Server) must first register with it, and the SPCS provides credentials for future usage. It is worth mentioning that the registration process is mainly accomplished over a private channel in offline mode, which is why open credentials don't affect the security of a scheme. The following steps will explain it comprehensively:

**Step 1:** The desired user first inputs their identity $ID_P$, and password $PW_P$ and generates biometrics $B_P$. The mobile device fetches $r_{P1}$ and calculates $Gen(B_P)=(\alpha_P, \beta_P)$, $HPW_P=h(ID_P||r_{P1}||PW_P)$ and sends $\{HPW_P, ID_P\}$ message towards the $S_{PCS}$.

**Step 2:** The $S_{PCS}$ confirms the patient's identity; if it exists, tell them the selection of another unique identity; else, calculates $Q_P=h(ID_P||s)$, $G_P=HPW_P \oplus Q_P$, retrieves $r_{PCS}$, computes $CID_P=E_s(ID_P||r_{PCS})$, $C_P=h(ID_P||Q_P||HPW_P)$, send and stores $\{CID_P, G_P, C_P, h(.)\}$ in the mobile-device of a patient where it calculates $V_P=r_P \oplus h(\alpha_P)$ and keeps $\{V_P, \beta_P\}$ parameters, as shown in Fig 2.

## 4.3 Authentication phase

**Step 1:** In this protocol's phase, a legitimate patient enters their identity $ID_P$, password $PW_P$, and generates $B_P$.

**Step 2:** The mobile device with the patient fetches a random number $r_{P2}$ and $G_P$ from memory, calculates $\alpha_P=Rep(B_P, \beta_P)$, $r_{P1}=r_{P2} \oplus h(\alpha_P)$, $HPW_P=h(ID_P||r_{P1}||PW_P)$, $Q_P=HPW_P \oplus G_P$, $C_P'=h(ID_P||Q_P||HPW_P)$ confirms $C_P'?=C_P$. If verified, the mobile device fetches another random number $a \in Z^*_q$ and calculates $W_P=a.O$, $V_P=h(ID_P||Q_P||W_P||T_1)$ and transmits $\{W_P, V_P, CID_P\}$ message towards $S_{PCS}$ over a public channel.

**Step 3:** The $S_{PCS}$ confirms the time and decrypts $CID_P$ using the same secret key of $S_{PCS}$, which is $(ID_P||r_{PCS})=D_s(CID_P)$, $Q_P=h(ID_P||s)$, $V_P'= h(ID_P||Q_P||W_P||T_1)$, confirms $V_P'?=V_P$, if holds, $S_{PCS}$ fetches another random number $b \in Z^*_q$, calculates $X_P=b.O$, $SK_{PCS}=h(ID_P||Q_P||X_P||T_1)$, create a new number $r_{PCS}^{new} \in Z^*_q$, $CID_P^{new}=E_s(ID_P||r_{PCS}^{new})$, $V_P=h(Q_P||CID_P^{new}||SK_{PCS}||T_2)$ and transmits $\{X_P, V_P, CID_P^{new}, T_2\}$ back towards patient over an open channel.

**Step 4:** The mobile device with the patient calculates $SK_P=h(ID_P||Q_P||W_P||T_1)$, $V_P=h(Q_P||CID_P^{new}||SK_P||T_2)$, confirms $V_P?=V_P$ and replaces $CID_P$ with $CID_P^{new}$, as shown in Fig 3.

## 4.4 Password change phase

In this phase, patients can freely and securely change their passwords and biometrics. In this regard, the patient/user first provides their old $ID_P$ and $PW_P$ and imprints $B_P$ in the application program installed on their mobile. It then calculates $\alpha_P=Rep(B_P, \beta_P)$, $r_{P1}=r_{P2} \oplus h(\alpha_P)$, $HPW_P=h(ID_P||r_{P1}||PW_P)$, and $Q_P=HPW_P \oplus G_P$. It verifies $C_P?=h(ID_P||Q_P||HPW_P$, if

| U (User) | PCS (Public Cloud Server) |
|---|---|
| Enter: $ID_P$, $PW_P$ | |
| Generate: $B_P$ | |
| Pick: $r_{P2}$ $G_P$ from the memory | |
| Calculate: $\alpha_P = Rep(B_P, \beta_P)$ | |
| $r_{P1} = r_{P2} \oplus h(\alpha_P)$ | |
| $HPW_P = h(ID_P \| r_{P1} \| PW_P)$ | |
| $Q_P = HPW_P \oplus G_P$ | |
| $C_P^{/} = h(ID_P \| Q_P \| HPW_P)$ | |
| Confirm: $C_P^{/} ?= C_P$ | |
| Pick: $a \in Z_q^*$ | |
| Calculate: $W_P = a.O$ | |
| $V_P = h(ID_P \| Q_P \| W_P \| T_1)$ | |

$$\xrightarrow{\{W_P, V_P, CID_P, T_1\}}$$

Confirm: $T_1$
Decrypt: $CID_P$
$(ID_P \| r_{PCS}) = D_s(CID_P)$
Compute: $Q_P = h(ID_P \| s)$
$V_P^{/} = h(ID_P \| Q_P \| W_P \| T_1)$
Confirm: $V_P^{/} ?= V_P$
Pick: $b \in Z_q^*$
Calculate: $X_P = b.O$
$SK_{PCS} = h(ID_P \| Q_P \| X_P \| T_1)$
Pick: $r_{PCS}^{new} \in Z_q^*$
Compute: $CID_P^{new} = E_s(ID_P \| r_{PCS}^{new})$
$V_P = h(Q_P \| CID_P^{new} \| SK_{PCS} \| T_2)$

$$\xleftarrow{\{X_P, V_P, CID_P^{new}, T_2\}}$$

Confirm: $T_2$
Calculate: $SK_P = h(ID_P \| Q_P \| W_P \| T_1)$
$V_P = h(Q_P \| CID_P^{new} \| SK_P \| T_2)$
Confirm $V_P ?= V_P$ and replace $CID_P$ with $CID_P^{new}$

**Fig 2. Registration Phase.**

| User (U) | Public Cloud Server (PCS) |
|---|---|
| Input $ID_P$, $PW_P$, | |
| Generate: $B_P$ and picks $r_{P1}$ | |
| Compute: $Gen(B_P) = (\alpha_P, \beta_P)$ | |
| $HPW_P = h(ID_P \| r_{P1} \| PW_P)$ | |

$$\xrightarrow{\{HPW_P, ID_P\}}$$

Confirm: $ID_P$
Compute: $Q_P = h(ID_P \| s)$
$G_P = HPW_P \oplus Q_P$
Retrieve: $r_{PCS}$
Compute: $CID_P = E_s(ID_P \| r_{PCS})$
$C_P = h(ID_P \| Q_P \| HPW_P)$

$$\xleftarrow{\{CID_P, G_P, C_P, h(.)\}}$$

$V_P = r_P \oplus h(\alpha_P)$
$\beta_P = Rep(\alpha_P, B_P)$ and keeps $\{V_P, \beta_P\}$

**Fig 3. Authentication Phase.**

confirmed, the patient will be asked to provide a new password $PW_P^{new}$ and biometrics $B_P^{new}$. The installed application program will generates $r_P^{new}$, computes $Gen(B_P^{new})=(\alpha_P^{new}, \beta_P^{new})$, $HPW_P^{new}=h(ID_P||r_P^{new}||PW_P^{new})$, $G_P^{new}=HPW_P^{new}\oplus Q_P$, $C_P^{new}=h(ID_P||Q_P||HPW_P^{new})$, $r_{P1}=r_{P2}\oplus h(\alpha_P^{new})$ and updates $\{r_{P1}, C_P, r_{P2}, \beta_P\}$ with $\{r_{P1}^{new}, C_P^{new}, r_{P2}^{new}, \beta_P^{new}\}$.

## 4.5 Patient revocation/re-registration phase

The proposed authentication protocol provides the facility to revoke/re-issue a patient to/from the public cloud server. In this regard, a patient first verifies their $ID_P$, $PW_P$, inputs biometrics $B_P$, and removes the random number $r_{P1}$ from the records. When the random numbers become removed from the information table and the patient tries to log in from their mobile device, the public cloud server rejects their request because $r_{P1}$ is not available in the record. Similarly, for the re-registration, the patient enters their identity and the public cloud server checks whether it is in the history along with status; if it exists and is in an inactive state, the whole registration phase is executed, the position becomes changed from passive to active by reactivating the patient to the system, as shown in the Fig 4 in the form of flowchart.

## 5 Security analysis

This section will check the protocol's security using different techniques like BAN logic [34] analysis, ROM [35] analysis, ProVerif [36] simulation, and threats analysis. These are discussed as under:

### 5.1 BAN logic analysis

BAN is a logic of belief was first introduced by three scientists and therefore named as Burrows–Abadi–Needham [34] which is purely used for checking random number reliability, trust, and accuracy in the protocol's participants. Different notations and rules used in BAN logic are expressed in Table 3, when checking the security of the proposed protocol, we will first express the different rules of BAN logic, define goals, represent idealization and confesses assumptions in achieving the specified goals. These foundations of BAN logic can be explained one by one as under:

**i. BAN Logic Rules.** BAN rules are comprehensively defined as under:

*Message Meaning*: Suppose Alice A believes in the communication between Alice A and Bob B via a key K and sees a message X combined with key K. In that case, Alice A also believes Bob B has jurisdiction over message X.

$$\frac{A| \equiv A\xleftrightarrow{K}B \triangleleft < X >_K}{A| \equiv B| \sim X} \tag{1}$$

*Verification*: If Alice A believes in the freshness of message X, Alice A believes Bob B once said X, then Alice A and Bob B think in message X.

$$\frac{A| \equiv A\xleftrightarrow{K}B \triangleleft < X >_K}{A| \equiv B| \sim X} \tag{2}$$

*Freshness*: If Alice A believes in the freshness of message X, Bob B also believes that message X is fresh.

$$\frac{A| \equiv (X)}{B| \equiv (X)} \tag{3}$$

*Jurisdiction*: If Alice A and Bob B believe the jurisdiction of message X and Alice A and Bob B

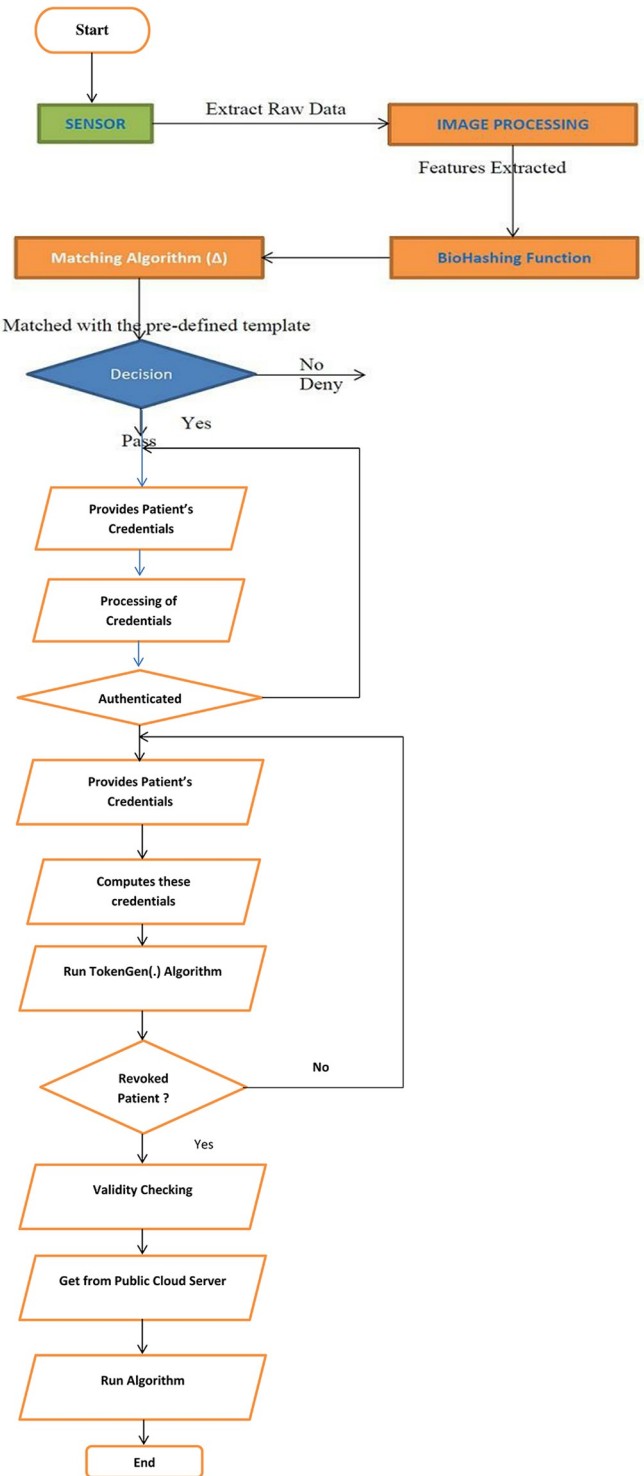

**Fig 4. Flow-chart of the whole scenarios.**

**Table 3. BAN notations and descriptions.**

| Notation | Description |
| --- | --- |
| A $\mid\equiv$ B | Alice A *believe*s in Bob B |
| A $\Rightarrow$ B | Alice A has *jurisdiction* over Bob B |
| A $\mid\sim$ B | Alice A *once said* Bob B |
| A $\lhd$ B | Alice A *sees* Bob B |
| #(X) | X is said to be *fresh* |
| $<$A$>_{\mathrm{L}}$ | Alice A message is *combined* with formula L |
| {A}$_{\mathrm{K}}$ | Alice A message is *encrypted* with key K |
| (A)$_{\mathrm{K}}$ | Alice A message is *hashed* with key K |

once said message X, then Alice A and Bob B think of message X.

$$\frac{A\mid \equiv (X), A\mid \equiv B\mid \sim X}{A\mid \equiv B) \equiv X} \tag{4}$$

**ii. BAN Logic Goals.** BAN logic goals for the proposed scheme are as under:

$$U\mid \equiv (U\xleftrightarrow{SK} PCS) \tag{5}$$

$$U\mid \equiv S\mid \equiv (U\xleftrightarrow{SK} PCS) \tag{6}$$

$$PCS\mid \equiv (U\xleftrightarrow{SK} PCS) \tag{7}$$

$$PCS\mid \equiv U\mid \equiv (U\xleftrightarrow{SK} PCS) \tag{8}$$

iii. BAN Logic Idealizations
BAN logic idealizations for the proposed scheme are as under:

$$U \rightarrow PCS : (IDP, aP) : (U\xleftrightarrow{h(ID_P||s||T_1)} PCS) \tag{9}$$

$$PCS \rightarrow U : (bP, U\xleftrightarrow{SK} PCS) : (U\xleftrightarrow{h(ID_P||s||T_1)} PCS) \tag{10}$$

iv. BAN Logic Assumptions
BAN logic assumptions for the proposed scheme are as under:

$$U\mid \equiv (aP) \tag{11}$$

$$PCS\mid \equiv (bP) \tag{12}$$

$$U\mid \equiv (U\xleftrightarrow{h(ID_P||s||T_1)} PCS) \tag{13}$$

$$PCS| \equiv (U \overset{h(ID_P||s||T_1)}{\longleftrightarrow} PCS) \tag{14}$$

$$U| \equiv PCS| \Rightarrow (U \overset{SK}{\longleftrightarrow} PCS) \tag{15}$$

$$PCS| \equiv U| \Rightarrow (U \overset{SK}{\longleftrightarrow} PCS) \tag{16}$$

v. BAN Logic Proof

BAN logic proof for the proposed scheme is as under:

According to Eq (9), we get

$$U \rightarrow PCS: (IDP, aP): (U \overset{h(ID_P||s||T_1)}{\longleftrightarrow} PCS) \tag{17}$$

$$PCS \triangleleft (IDP, aP): (U \overset{h(ID_P||s||T_1)}{\longleftrightarrow} PCS) \tag{18}$$

As per Eq (14), we get

$$PCS| \equiv U| \sim (IDP, aP)(U \overset{h(ID_P||s||T_1)}{\longleftrightarrow} PCS) \tag{19}$$

$$PCS| \equiv U| \sim (IDP, aP)(U \overset{SK}{\longleftrightarrow} PCS) \tag{20}$$

Looking into Eqs (12) and (2), we get

$$PCS| \equiv U| \equiv (IDP, aP)(U \overset{SK}{\longleftrightarrow} PCS) \tag{21}$$

$$PCS| \equiv U| \equiv (U \overset{SK}{\longleftrightarrow} PCS) \tag{22}$$

G3 Achieved

As per Eq (22), we get

$$PCS| \equiv (U \overset{SK}{\longleftrightarrow} PCS) \tag{23}$$

G4 Achieved

Now, taking Eq (10), we get

$$PCS \rightarrow U: (bP, U \overset{SK}{\longleftrightarrow} PCS): (U \overset{h(ID_P||s||T_1)}{\longleftrightarrow} PCS) \tag{24}$$

$$U| \equiv (bP, U \overset{SK}{\longleftrightarrow} PCS): (U \overset{h(ID_P||s||T_1)}{\longleftrightarrow} PCS) \tag{25}$$

According to Eqs (24) and (13), we get

$$U| \equiv ((bP, U \overset{SK}{\longleftrightarrow} PCS): (U \overset{h(ID_P||s||T_1)}{\longleftrightarrow} PCS) \tag{26}$$

$$U| \equiv PCS| \sim (bP, U \overset{SK}{\longleftrightarrow} PCS) \tag{27}$$

Eqs ([9]) and ([26]), we get

$$U| \equiv PCS| \equiv (U \xleftrightarrow{SK} PCS) \tag{28}$$

G2 Achieved

Eq ([28]) can also be written as:

$$U| \equiv (U \xleftrightarrow{SK} PCS) \tag{29}$$

G1 Achieved

BAN logic goals for the proposed scheme are as under:

$$U| \equiv (U \xleftrightarrow{SK} PCS) \tag{5}$$

$$U| \equiv S| \equiv (U \xleftrightarrow{SK} PCS) \tag{6}$$

$$PCS| \equiv (U \xleftrightarrow{SK} PCS) \tag{7}$$

$$PCS| \equiv U| \equiv (U \xleftrightarrow{SK} PCS) \tag{8}$$

**iii. BAN Logic Idealizations.** BAN logic idealizations for the proposed scheme are as under:

$$U \to PCS : (ID_P, aP) : (U \xleftrightarrow{h(ID_P||s||T_1)} PCS) \tag{9}$$

$$PCS \to U : (bP, U \xleftrightarrow{SK} PCS) : (U \xleftrightarrow{h(ID_p||s))T_1)} PCS) \tag{10}$$

**iv. BAN Logic Assumptions.** BAN logic assumptions for the proposed scheme are as under:

$$U| \equiv (aP) \tag{11}$$

$$PCS| \equiv (bP) \tag{12}$$

$$U| \equiv (U \xleftrightarrow{h(ID_p||s||T_1)} PCS) \tag{13}$$

$$PCS| \equiv (U \xleftrightarrow{h(ID_p||s||T_1)} PCS) \tag{14}$$

$$U| \equiv PCS| \Rightarrow (U \xleftrightarrow{SK} PCS) \tag{15}$$

$$PCS| \equiv U| \Rightarrow (U \xleftrightarrow{SK} PCS) \tag{16}$$

**v. BAN Logic Proof.** BAN logic proof for the proposed scheme is as under:

According to Eq (9), we get

$$U \to PCS : (ID_P, aP) : (U \stackrel{h(ID_P||s||T_1)}{\longleftrightarrow} PCS) \tag{17}$$

$$PCS \triangleleft (ID_P, aP) : (U \stackrel{h(ID_P||s||T_1)}{\longleftrightarrow} PCS) \tag{18}$$

As per Eq (14), we get

$$PCS| \equiv U| \sim (ID_P, aP)(U \stackrel{h(ID_P||s||T_1)}{\longleftrightarrow} PCS) \tag{19}$$

$$PCS| \equiv U| \sim (ID_P, aP)(U \stackrel{SK}{\longleftrightarrow} PCS) \tag{20}$$

Looking into Eqs (12) and (2), we get

$$PCS| \equiv U| \equiv (ID_P, aP)(U \stackrel{SK}{\longleftrightarrow} PCS) \tag{21}$$

$$PCS| \equiv U| \equiv (U \stackrel{SK}{\longleftrightarrow} PCS) \tag{22}$$

$G_3$ Achieved
As per Eq (22), we get

$$PCS| \equiv (U \stackrel{SK}{\longleftrightarrow} PCS) \tag{23}$$

$G_4$ Achieved
Now, taking Eq (10), we get

$$PCS \to U : (bP, U \stackrel{SK}{\longleftrightarrow} PCS) : (U \stackrel{h(ID_P||s||T_1)}{\longleftrightarrow} PCS) \tag{24}$$

$$U| \equiv (bP, U \stackrel{SK}{\longleftrightarrow} PCS) : (U \stackrel{h(ID_P||s||T_1)}{\longleftrightarrow} PCS) \tag{25}$$

According to Eqs (24) and (13), we get

$$U| \equiv ((bP, U \stackrel{SK}{\longleftrightarrow} PCS) : (U \stackrel{h(ID_P||s||T_1)}{\longleftrightarrow} PCS) \tag{26}$$

$$U| \equiv PCS| \sim (bP, U \stackrel{SK}{\longleftrightarrow} PCS) \tag{27}$$

Eqs (9) and (26), we get

$$U| \equiv PCS| \equiv (U \stackrel{SK}{\longleftrightarrow} PCS) \tag{28}$$

$G_2$ Achieved
Eq (28) can also be written as:

$$U| \equiv (U \stackrel{SK}{\longleftrightarrow} PCS) \tag{29}$$

$G_1$ Achieved

## 5.2 ROM analysis

In this subsection of the article, we will construct some formal security analysis using Random Oracle Model (ROM) [35] discussions for scrutinizing the ECC key, secret key, and SHA-2

code security. Let adversary $A$ breaks the $F^{SK}$ using the following techniques, whereas $F^{SK}$ represents the function session key SK of the user side key computed for establishing a secure session [35].

*Game 1*: In the first step, the adversary attempts to access the public parameters of server $\lambda$ and is denoted by $G_1$ ($\lambda$, A), and the answer received is according to like a natural protocol output. $Win_0$ represents the winning chance with the adversary.

*Game 2*: The adversary interacts with the protocol by choosing a key $K_P$ and updating $F^{SK\text{-}key}$. The probability of winning this game with the adversary is $Win_1$ for colliding two ECC keys is:

$$|Prob_A[Win_1(F^{SK-ECC})] - Prob_A[Win_0]| \leq \frac{1}{2^{P(\lambda)}} \tag{30}$$

*Game 3*: In this game, the key for Enc(.) function is $k_1$, the key for the SHA-2 tag is $k_2$, and the key SHA-2 tag with reference is checked. Let advantage ADV with A is shown as:

$$|Prob_A[Win_2(F^{SK-ECC})] - [Prob_A[Win_1]| \leq 3ADV_{A^{P(\lambda)}}^{KF} \tag{31}$$

Where KF means key freshness

*Game 4*: In this game, the adversary launches a forgery attack on message m along with the SHA-2 tag and finds a fresh message/SHA-2 tag. The advantage with A to succeed is given as follows:

$$|Prob_A[Win_3(F^{SK-ECC})] - [Prob_A[Win_2]| \leq ADV_{A^{(SHA-2)^\lambda}}^{FM} \tag{32}$$

Where FM means forging a message

*Game 5*: in this game, the Enc(.)/Dec(.) functions made are replaced by an $A$ using the $k_1$ key and x (the personal values). The advantage with $A$ to win this game is:

$$|Prob_A[Win_4(F^{SK-ECC})] - [Prob_A[Win_3]| \leq ADV_{A^{Enc(.)/Dec(.)^\lambda}}^{Ind} \tag{33}$$

Where Ind means in-distinguishability

*Game 6*: In this game, the adversary can attempt for the secret key $s$ chosen by PCS before executing the protocol. The probability with $A$ of guessing the accurate s is:

$$|Prob_A[Win_5(F^{SK-s})] - [Prob_A[Win_4]| \leq \frac{1}{2^{P(\lambda)}} \tag{34}$$

*Game 7*: The secret key $s_1$ for Enc(.), $s_2$ for SHA-2 tag, and s is for execution, then the advantage with $A$ is:

$$|Prob_A[Win_6(F^{SK-s})] - [Prob_A[Win_5]| \leq 2ADV_{A^{P(\lambda)}}^{KF} \tag{35}$$

*Game 8*: By launching of forgery attack on secret credentials is performed in this step, and the advantage of winning the game is:

$$|Prob_A[Win_7(F^{SK-s})] - [Prob_A[Win_6]| \leq ADV_{A^{(r)^\lambda}}^{FM} \tag{36}$$

Where r means key randomness and FM means forging personal values.

*Game 9*: By using the secret values, we get

$$\left| Prob_A[Win_8(F^{SK-s})] - \left[ Prob_A[Win_7] \right| \leq ADV^{ID}_{A^{\overline{Dec(.)}}_{Enc(.)^\lambda}} \right. \tag{37}$$

$$|Prob_A[Win_8] \leq \begin{cases} 0 \\ 2^{-p^\lambda} \end{cases} \tag{38}$$

The final chance with *A* of winning the games can be calculated from these games is shown in Eq (39):

$$ADV_A^{SK-ECC}(\lambda, \text{A}) \leq 2\frac{1}{2^{P(\lambda)}} + 5ADV^{SK-s}_{A^{P(\lambda)}} + 2ADV^{SK-ECC}_{A^{(SHA-2)^\lambda}} + 2ADV^{SK-ss}_{A^{(SHA-2)^\lambda}} \tag{39}$$

Therefore, considering the above analysis, it has been clear that the in-distinguishability, freshness, and randomness in FSK are verified from Game1 to Game7 [Eqs (30–36), while the confidentiality security feature of the secret values is confirmed from Game8-Game9 [Eqs (37–39)].

## 5.3 ProVerif2.03 simulation

Another method in formal security proof used is to simulate the proposed protocol; in this regard, we have programmed by using a software verification toolkit ProVerif [36]. The ProVerif simulation is a world widely used toolkit for verifying the session key reachability, secrecy, integrity, and confidentiality. Upon running the code, the result shows that the protocol securely exchanges the secret session key among all the participants, and its integrity and reachability have been verified. The result is shown below:

## 5.4 Threats analysis

This subsection will scrutinize the scheme's security by considering prominent attacks like impersonation, online password guessing, stolen-verifier, replay, and offline password guessing attacks. These are described one by one as under:

| Completing equations... |
| --- |
| – Process 1– Query not attacker(SK[]) in process 1 |
| Starting query not attacker(SK[]) |
| RESULT not attacker(SK[]) is true. |
| Verification summary: |
| Query not attacker(SK[]) is true. |
| Query inj-event(end_PCS(IDPCS)) ==> inj-event(start_PCS(IDP)) is true. |

**a) Impersonation Attack.** The masquerade of a legitimate user logging into a server to avail valuable services, however, in the proposed scheme, let's suppose someone desires to log in, but they cannot because the procedure is protected in the login by a fuzzy extractor having Gen(.) and Rep(.) functions of the user's biometrics. Also, the 160-bit long ECC key provides security against illegitimate login attempts. The malicious user cannot enter the server without knowing the password, 160-bit ECC key, and other user credentials. Therefore, an impersonation attack cannot be valid on the proposed scheme.

**b) Online Password Guessing Attack.** If adversary $A$ attempts to guess the password by several tries, then A must know $r_1$ in the first round to successfully log in to the server. But doing so, $A$ cannot guess such a big 160-bit random ECC key; the probability with A is very low for guessing the exact ECC key. Similarly, in the proposed scheme, the password is not retained at the registration phase to be used later for login purposes. The user can set a password and transmit it to the server. If the server observes that someone attempted to enter another password, the server promptly sends a deny message, and the process is discarded. Therefore, the proposed protocol is much safe against online password-guessing attacks.

**c) Stolen Verifier Attack.** The server authenticates anyone from its secret key $s$ because no database is there to store the user credentials, so any attempt of an adversary fails. Therefore, the proposed scheme is safe against stolen verifier vulnerability.

**d) Replay attack.** Our scheme will discard an adversary's potential attempt to replicate an old message due to random checks at each round trip and time threshold. Therefore, this drawback needs to be revised needs to be modified in our scheme.

**e) Offline Password Guessing Attack.** Suppose an attacker obtains the share secrets like $ID_P$ $r_0$, $r_1$, $r_2$, and hash codes and later desires to launch an attack on the system by guessing the password in offline mode. However, A cannot succeed due to any knowledge of biometrics, and the fuzzy extractor makes it more unique, so their guessed information cannot compare with the valid parameters. Therefore, the said attempt doesn't exist in our scheme.

**f) Forgery Attack.** Due to unique session secret key for each session, generation of 160-bit random number from a curve and the presence of timestamp can guarantee forgery attack

**g) Known session key Attack.** The session key is computed from random numbers, timestamp, identities, biometric with fuzzy extractor, no one can launch attack on it. Also if an attacker enter the server and copy the previous session key, he/she cannot identify any credentials from it, as it is computed from different credentials which have been extracted randomly for session key computation.

**h) Man-in-the-Middle (MITM) Attack.** Suppose an adversary $A$ intercepts the public channel which the patient communicate his/her information to the cloud server and replaces/modifies/delete/updates the message, in the proposed protocol A cannot calculate $V_P=h(ID_P||Q_P||W_P||T_1)$ which is obtained from $Q_P=HPW_P\oplus G_P$. Without knowing $Q_P=HPW_P\oplus G_P$, $HPW_P$ and $G_P$ adversary cannot reached for server key $s$. Similarly, it is also impossible for A to exactly computes $V_P=h(Q_P||CID_P^{new}||SK_{PCS}||T_2)$ because A doesn't know $Q_P$ and other necessary credentials. Also for launching MIMT attack, A must passed $V_P^{/}=h(ID_P||Q_P||W_P||T_1)$ which is computationally infeasible for him to succeeded for the validation of patient and cloud server. Therefore, our security mechanism is must safe against MIMT attack.

**i) Privileged Insider Attack.** In the registration phase of the proposed protocol, the password sent is $HPW_P=h(ID_P||r_{P1}||PW_P)$ in which a privileged user cannot identify because $r_{p1}$ is 160-bit large random point of a curve, unknown to the server as well as the user therefore, our scheme is safe against privileged insider attack.

**j) DoS Attack.** As each round trip of the protocol consisted of a pre-defined time threshold, random checks and 160-bit large random numbers, the DoS attack did not succeed for our scheme. Because when an attacker can copy a message from the open channel, he/she cannot figure out proper credentials from it because the same message in the upcoming round is entirely different due to strong biometrics with fuzzy extractor, ECC key and timestamp. Therefore, our scheme is safe against DoS attacks. Any illegal attempt can automatically be captured due to random checks at each proposed security mechanism round trip.

**Table 4. Storage overheads analysis.**

| Participant | Parameters | Values in Bits | Total Overheads in Bits |
|---|---|---|---|
| User (U) | $ID_P$, $PW_P$, Gen(.), $V_P$, $\beta_P$, r | 64+32+128+416+256+160 | 1056 |
| Server (PCS) | O, q, E(Fp). h(.), s, $r_{PCS}$ | 160+160+160+256+256+160 | 1152 |
| | | **G/Total Storage Costs** | **2208 Bits** |

## 6 Performance analysis

In this section, the performance and comparison analysis of the proposed scheme can be measured by considering storage overheads, communication, and computation costs analysis. These performance metrics are described one by one as under:

### 6.1 Storage overheads analysis

The parameters stored in the registration phase of the proposed scheme are used to calculate the storage overheads of this scheme. So, according to JPBC2.0.0 [37], a 64-bit operating system namely Windows 10, core $i_5$ CPU having 8GB of RAM, according to [37, 38], the computation cost of ECC key is 160-bits, hash functions 256-bits, biometric Gen (.) & Rep (.) functions occupy 128-bit space, encryption/decryption functions 192-bits; then the storage costs/overheads of the proposed scheme are shown in Table 4.

### 6.2 Communication costs

The message exchanged during the authentication phase of the scheme is counted to be the communication costs of a protocol; keeping in view [39], the said costs/overheads of the proposed method are shown in Table 5.

### 6.3 Computation costs

The time consumed while performing different operations of computation is the computation cost. According to [39], the proposed scheme, let's suppose $T_H$ represents the time of hash function, $T_{ECC}$ describes the extraction of the ECC key, $T_{XOR}$ is the XOR computation time, and $T_{E/D}$ is the time required to compute encryption/decryption functions. The overall computation costs for the proposed scheme are shown in Table 6. It is to mention that the computation costs are considered only for the authentication phase of any authentication scheme; therefore, according to [39], suppose $T_{ECC}$ is 19.2 ms, $T_H$ is 0.32 ms, $T_{XOR}$ is negligible equal to zero, and $T_{E/D}$ is 5.6 ms, then the computation costs for the proposed scheme is given as under:

### 6.4 Comparison analysis

In this sub-section, we comparatively analyze the proposed protocol regarding the security goals defined in section 2 of the paper with [40–47], as shown in Table 7. Similarly, the

**Table 5. Communication costs analysis.**

| Participants | Message | Values in Bits | Total Costs in Bits |
|---|---|---|---|
| U→PCS | {$W_P$, $V_P$, $CID_P$, $T_1$} | 256+160+192+56 | 664 |
| PCS→U | {$X_P$, $V_P$, $CID_P^{new}$, $T_2$} | 160+256+192+56 | 664 |
| | | **G/Total Communication Costs** | |

**Table 6. Computation costs analysis.**

| Phase | Peer | Operations | Total |
|---|---|---|---|
| Registration | U | $1T_{ECC}+2T_H+1T_{XOR}+0T_{E/D}$ | $3T_{ECC}+10T_H+1T_{XOR}+3T_{E/D}$ |
| | PCS | $1T_{ECC}+2T_H+1T_{XOR}+1T_{E/D}$ | $3(19.2)+10(0.32)+0+3(5.6)$ |
| Authentication | U | $1T_{ECC}+6T_H+1T_{XOR}+1T_{E/D}$ | $57.6+3.2+16.8$ |
| | PCS | $2T_{ECC}+4T_H+0T_{XOR}+2T_{E/D}$ | $77.6$ ms |
| Password Change | U | $2T_{ECC}+3T_H+4T_{XOR}+0T_{E/D}$ | |
| | PCS | $2T_{ECC}+5T_H+3T_{XOR}+0T_{E/D}$ | |

**Table 7. Comparison analysis in terms of communication costs.**

| Schemes Performance Metrics↓ | [40] | [41] | [42] | [43] | [44] | [45] | [46] | [47] | Proposed |
|---|---|---|---|---|---|---|---|---|---|
| Communication Costs in Bits | 2240 | 1996 | 2196 | 1792 | 5856 | 1360 | 2272 | 2088 | 1328 |
| Computation Costs in ms | 290.00 | 188.11 | 101.05 | 178.10 | 111.35 | 163.84 | 080.88 | 091.12 | 077.60 |

proposed scheme can also be compared with [40–47] in terms of communication costs and computation costs are shown in Figs 5 and 6, so that to check the balance of security with performance, both are necessary to be balanced. If one of them is improved, and the other does not, it means that the work couldn't implement practically for the said sensitive environment, as shown in Table 8 below:

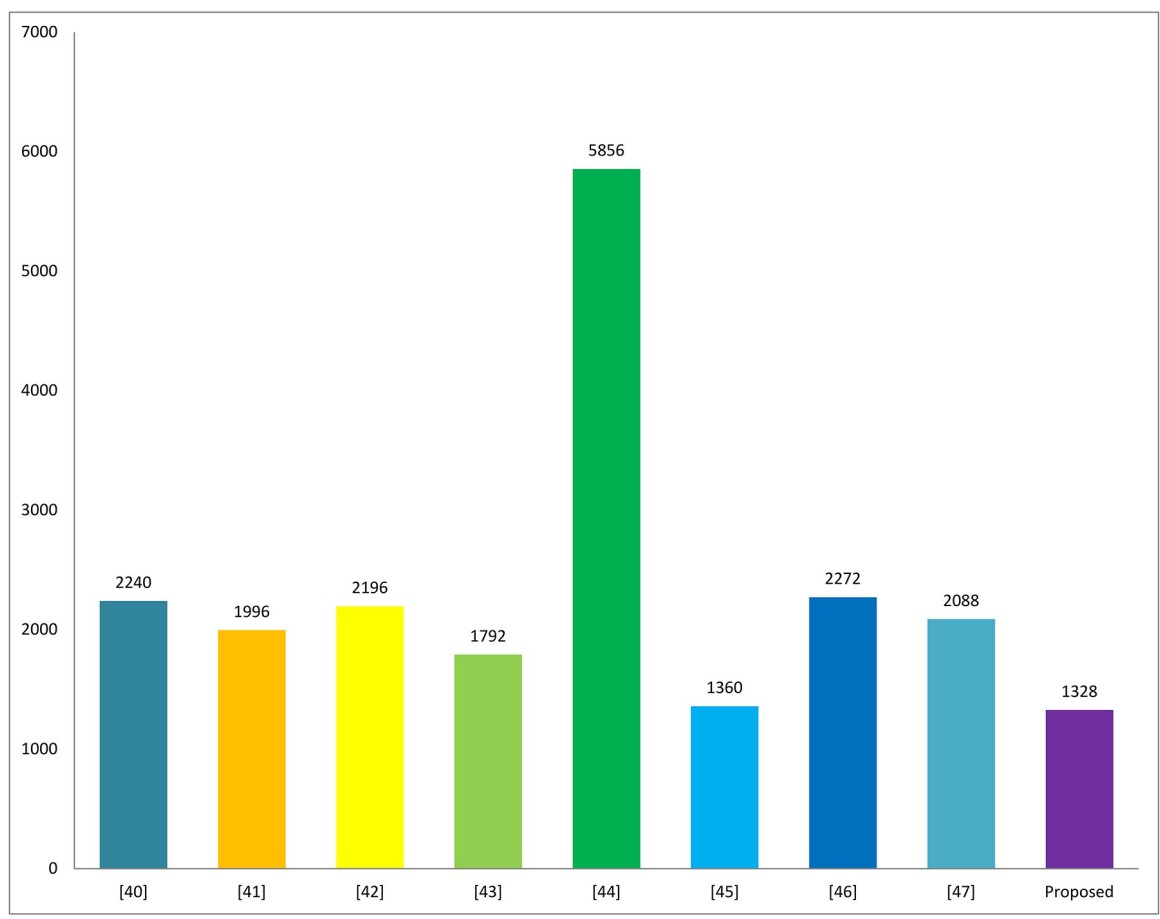

**Fig 5. Communication costs comparison.**

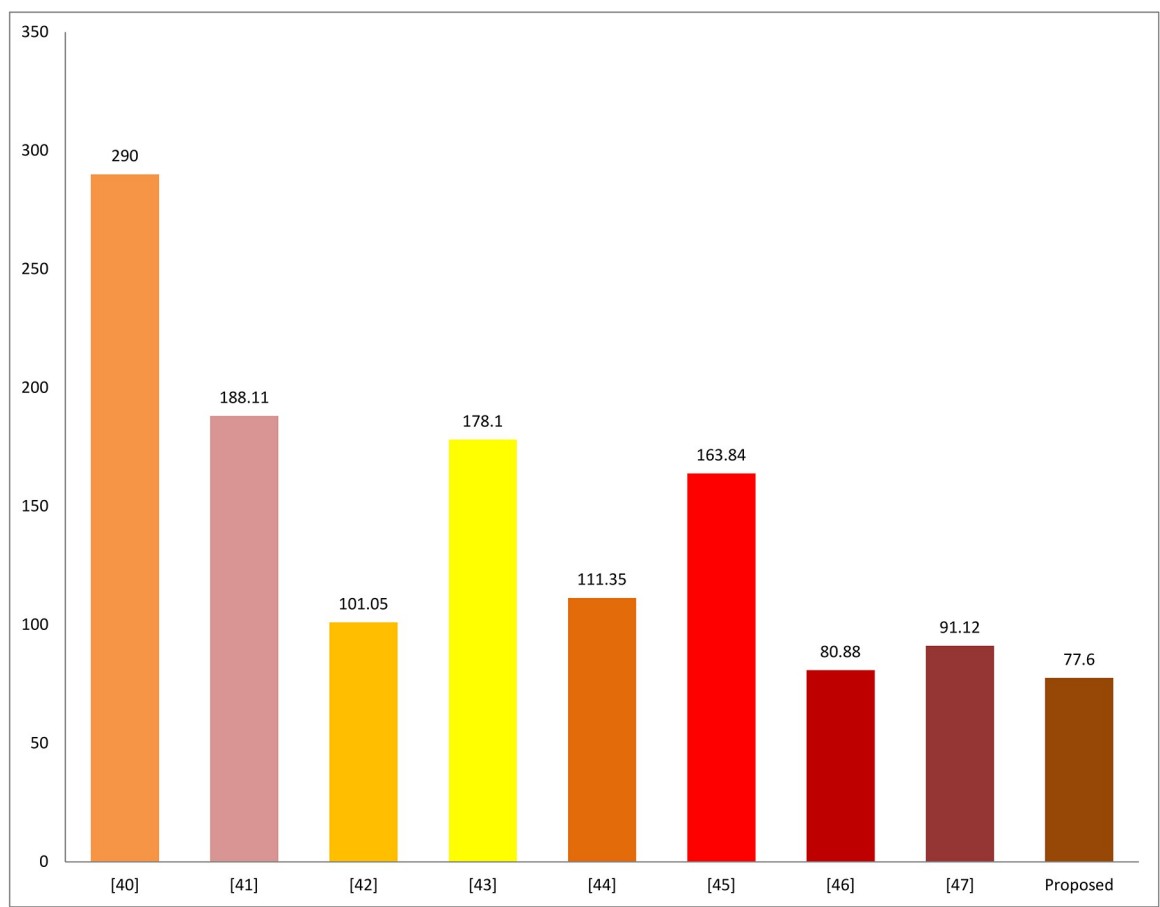

**Fig 6. Computation costs comparison.**

Keeping in view the aforementioned percentage difference (Table 9), our scheme is 40.71% better than [40], 33.46% than [41], 39.52% than [42], 77.32% than [43], 2.38% than [45], 41.54% than [46], and 36.39% than [47]. Compared to all, our scheme is 33.91% better in communication costs.

**Table 8. Comparison analysis in terms of security features.**

| Goal | Description | [40] | [41] | [42] | [43] | [44] | [45] | [46] | [47] | Proposed |
|------|-------------|------|------|------|------|------|------|------|------|----------|
| Goal1 | De-synchronization attack | N | N | N | N | Y | N | N | Y | N |
| Goal2 | Mutual Authentication | Y | Y | Y | Y | Y | Y | Y | Y | Y |
| Goal3 | spoofing attack | N | Y | N | N | N | N | Y | N | N |
| Goal4 | insider attack | N | N | Y | Y | Y | N | N | N | N |
| Goal5 | Prevention of false messaging | Y | Y | N | Y | Y | Y | Y | Y | Y |
| Goal6 | Anonymity | N | Y | Y | Y | Y | N | Y | Y | Y |
| Goal7 | Accountability | N | N | Y | N | N | N | N | Y | Y |
| Goal8 | replay attack | N | N | N | N | N | N | N | N | N |
| Goal9 | Impersonation attack | N | N | N | Y | N | N | N | N | N |
| Goal10 | perfect forward secrecy | Y | Y | Y | N | Y | Y | Y | Y | Y |
| Goal11 | un-traceability | N | Y | Y | Y | N | N | Y | N | N |
| Goal12 | ESL Attack | N | Y | Y | N | N | N | Y | N | N |

**Table 9. Percentage (%) improvement in performance metrics of the proposed protocol.**

The percentage difference between communication and computation costs $= \left( \frac{\text{Comm.Cost} - \text{Comp.Cost}}{\frac{(\text{Comm.Cost} + \text{Comp.Cost})}{2}} \right) \times 100$

| | |
|---|---|
| $= \left( \frac{(1360-1328)}{\frac{(1360+1328)}{2}} \right) \times 100$ | $= \left( \frac{(80.88-77.60)}{\frac{(80.88+77.60)}{2}} \right) \times 100$ |
| $= \left( \frac{32}{\frac{2688}{2}} \right) \times 100$ | $= \left( \frac{3.28}{\frac{2688}{2}} \right) \times 100$ |
| $= \left( \frac{32 \times 2}{2688} \right) \times 100$ | $= \left( \frac{3.28 \times 2}{158.48} \right) \times 100$ |
| $= \left( \frac{64}{2688} \right) \times 100$ | $= \left( \frac{6.56}{158.48} \right) \times 100$ |

= 2.38% = 4.13%

Which means that our scheme is 2.38% smaller (lightweight) Which means that our scheme is 4.13% is speedy

in size in term of communication overheads. in computation than their competitors.

Keeping in view

Similarly, for computation costs our scheme is 73.24% superior in computation than [40], 58.74% than [41], 23.20% than [42], 56.42% than [43], 30.30% than [44], 52.63% than [45], 4.13% than [46] and 14.83% than [47]. In average, compared to all, in terms of computation, our scheme takes 35.39% less CPU time than its competitors.

## 7. Conclusion

The patient-sensitive information broadcasting over a public network channel is unsafe as the wireless channels are exposed to various security threats and need protection from multiple attacks. Such a susceptible environment cannot be protected without proper authentication of the end-user, cloud server, and data stored in the server obviously needs a lightweight, robust, and efficient authentication scheme to secure the stored credential and confirm end-user privacy. Therefore, this work provides a method/authentication scheme based on ECC using fuzzy extractor methods. The biometrics generated by the end-users have the capabilities of minimum loss of character and perfect uniqueness due to the fuzzy extractor method. The security of the proposed scheme has been tackled using BAN logic, ROM analysis, Pro-Verif2.03 simulation, and attack analysis. While in the performance analysis section, we have considered storage, computation, and communication overheads. The result in the comparative analysis section shows that the proposed security mechanism is lightweight and robust to its competitors and can be recommended for a real-world cloud-based healthcare system.

In the future, we plan to develop a secure system using the Cyber Shave Chaotic map method – a double-layer security method in which we will use cryptography in the first layer and steganography in the second layer to protect patient-sensitive information.

## Supporting information

**S1 Appendix. ProVerif2.03 code.**
(TXT)

**S1 File. Figures.**
(PDF)

**S2 File. Author biography.**
(DOCX)

## Acknowledgments

The authors extend their appreciation to the Deanship of Scientific Research at University of Bisha for supporting this research through the general research project under grant number (UB-GRP- 65 -1444) and through the Fast Track Research Support Program. The authors are also thankful to Faculty of Computer Science and Information technology, Universiti Malaysia Sarawak, Malaysia for their support.

## Author Contributions

**Conceptualization:** Irshad Ahmed Abbasi, Saeed Ullah Jan.

**Data curation:** Irshad Ahmed Abbasi, Saeed Ullah Jan, Abdulrahman Saad Alqahtani, Fahad Algarni.

**Formal analysis:** Irshad Ahmed Abbasi, Saeed Ullah Jan, Fahad Algarni.

**Funding acquisition:** Irshad Ahmed Abbasi, Saeed Ullah Jan, Abdulrahman Saad Alqahtani, Adnan Shahid Khan, Fahad Algarni.

**Investigation:** Irshad Ahmed Abbasi, Saeed Ullah Jan, Adnan Shahid Khan, Fahad Algarni.

**Methodology:** Irshad Ahmed Abbasi, Abdulrahman Saad Alqahtani, Adnan Shahid Khan.

**Project administration:** Irshad Ahmed Abbasi, Saeed Ullah Jan, Fahad Algarni.

**Resources:** Irshad Ahmed Abbasi, Abdulrahman Saad Alqahtani, Adnan Shahid Khan, Fahad Algarni.

**Software:** Irshad Ahmed Abbasi, Saeed Ullah Jan, Abdulrahman Saad Alqahtani.

**Supervision:** Irshad Ahmed Abbasi, Fahad Algarni.

**Validation:** Irshad Ahmed Abbasi, Abdulrahman Saad Alqahtani, Adnan Shahid Khan, Fahad Algarni.

**Visualization:** Irshad Ahmed Abbasi, Saeed Ullah Jan, Adnan Shahid Khan.

**Writing – original draft:** Irshad Ahmed Abbasi.

**Writing – review & editing:** Irshad Ahmed Abbasi, Saeed Ullah Jan, Abdulrahman Saad Alqahtani, Adnan Shahid Khan, Fahad Algarni.

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
