## [Decision Letter · Decision Letter 0]

12 Oct 2023

PONE-D-23-30535A Lightweight and Robust Authentication Scheme for the Healthcare System using Public Cloud ServerPLOS ONE

Dear Dr. Abbasi,

Thank you for submitting your manuscript to PLOS ONE. After careful consideration, we feel that it has merit but does not fully meet PLOS ONE’s publication criteria as it currently stands. Therefore, we invite you to submit a revised version of the manuscript that addresses the points raised during the review process.

We look forward to receiving your revised manuscript.

Kind regards,

Vincent Omollo Nyangaresi, Ph.D

Academic Editor

PLOS ONE

Journal Requirements:

Additional Editor Comments:

1) This paper has numerous sentence structure issues that need to be revised. For example:

"...Despite the many advantages of cloud computing technologies for the healthcare industry, still, a variety of management, technological control, and regulatory issues are also to be taken into consideration, but two main issues, i.e., privacy and security of the current health system, are challenging for academia and industries..."

Consider breaking down the above sentence in two or more sentences for ease of understanding.

2) The following sentence is incomplete:

"..Because the healthcare system is gradually being converted to the public cloud environment in offering services to patients, researchers, labs, hospitals etc., in a flexible manner with minimal operational costs and in a reliable manner...."

3) In the abstract, you need to describe the current solutions to public cloud server security issues. Thereafter, explain the challenges of these current solutions. This will help readers see the gap that this paper sought to fill.

4) Towards the end of the abstract, give a summary of the obtained results, preferably in terms of percentage improvements.

5) "..A fuzzy extractor method is used to design the proposed protocol to make the authentication process more secure, and the system shows uniqueness while performing any task...."

i) Which task are you referring to?

ii) Which is this uniqueness that you are insinuating?

6)"...The proposed protocol is verifiably secure; this article has proved it using a well-known technique, i.e., BAN logic...."

You may state the above as follows:

"The BAN logic and ProVerif security proofs carried out demonstrate that the proposed protocol is verifiably secure"

7)In the Literature Survey section, provide a critique of each of the current works discussed. This will help the readers appreciate the contributions of the proposed protocol.

8)In "Review Analysis of Baseline Scheme", replace 'Baseline scheme' with the actual scheme that you are referring to. You may say, "Review Analysis of Azrour et al.'s scheme". The same should be done to "Weaknesses in Baseline Scheme"

9)In section "Formal Security Analysis through Pragmatic Illustration", replace 'Pragmatic Illustration' with 'Real or Random model'

10) In Section 5.4, "Informal Security Analysis through Discussion", remove 'through Discussion'

11)Replace, "The parameters stored in the registration phase of the proposed scheme are the storage overheads" with "The parameters stored in the registration phase of the proposed scheme are used to calculate the storage overheads of this scheme"

12) Include some future research scope in the conclusion section.

Reviewers' comments:

Reviewer's Responses to Questions

**Comments to the Author**

1. Is the manuscript technically sound, and do the data support the conclusions?

Reviewer #1: Yes

Reviewer #2: Partly

2. Has the statistical analysis been performed appropriately and rigorously? 

Reviewer #1: Yes

Reviewer #2: Yes

3. Have the authors made all data underlying the findings in their manuscript fully available?

Reviewer #1: Yes

Reviewer #2: Yes

4. Is the manuscript presented in an intelligible fashion and written in standard English?

Reviewer #1: Yes

Reviewer #2: Yes

5. Review Comments to the Author

Reviewer #1: 1- Is the following point considered one of the researcher’s contributions?: "An ECC-based lightweight authentication protocol has been proposed to securely provide services to the end-user in the cloud-enabled healthcare system"

2- The references used in Literature Survey are not arranged from oldest to newest. In addition to the limitations that some references suffered from, listed in Table 1, was the researcher able to overcome them? Please explain this in a separate paragraph within the section listed in Table 1

3-The references listed in Tables 7 and 8 are not included in the list of related works in Table 1.

4- It is possible to use some numerical values for the communications costs of the proposed method in the conclusions. In addition to giving a future outlook for upcoming works in the same field

5- Standardize the format of references. In addition to including at least two references published during the years 2022 and 2023

Reviewer #2: Authors proposed low-complexity, and secure authentication healthcare system using public cloud serve. Here are some comments:

1- In abstract, why low-complexity operations are required, the purpose has not been clarified?

2- In abstract, it is not clear whether the security process is for authentication or for storing data in the cloud server.

3- In abstract, the limitations of previous work were not addressed.

4- The introduction lacks references.

5-The most closely related works obstacles to the proposed work were not explained in the introduction, nor was the mechanism of action of the proposed method to overcome them explained.

6- Novelty is tinged with confusion.

7-The “Related Work” section lacks of enough references. I strongly recommend that the author improve this section by adding references that support all the claims and motivation of the problem. The author may precisely and comprehensively point out the current issues and existing solutions. I suggest adding more related reference such as:

1- https://ieeexplore.ieee.org/abstract/document/9842900

2- https://ieeexplore.ieee.org/abstract/document/9842900

3- https://ieeexplore.ieee.org/abstract/document/7753621

4- https://www.mdpi.com/2076-3417/13/2/691

5- https://ieeexplore.ieee.org/abstract/document/9274321

8- In related works section, the limitations of most works must be addressed and proven with a scientific argument or citation to a reference.

9- All works within Table 1 must be included in the text of related works.

10- In “3.2 Elliptic Curve Cryptography (ECC)”, the last point is not clear.

11- In 3.4 “Threat and Adversary Model”, the seventh point requires elaboration.

12- In “Weaknesses in Baseline Scheme”, 4) Privileged-insider Attack, Proof is not enough.

13-Identity in the registration process is transmitted openly and explicitly. This is a safe loophole.

14- In “4.5 Patient Revocation/Re-registration Phase”, add flowchart explain this process.

15- In security analysis, many attacks are not addressed, such as: forgery, known session key, Man-in-the-middle, replay, impersonation, spoofing, eavesdropping and ephemeral secret leakage, KSSTI, Privileged insider and guessing, de-synchronization and DOS attacks.

16- The attacks in Table 8 must be security-analyzed and not just referred to in the table.

6. PLOS authors have the option to publish the peer review history of their article (what does this mean?). If published, this will include your full peer review and any attached files.

Reviewer #1: No

Reviewer #2: **Yes: **Zaid Ameen Abduljabbar

---

## [Author Response · Author response to Decision Letter 0]

20 Oct 2023

Editor: Dear Honorable Editor, we have addressed all the comments, many thanks for your valuable concerns as it has improved our manuscript.

Reviewer 1: Dear Reviewer, we have incorporated all of your suggestions or concerns into our revision. They were very helpful. Thanks for your help in improving our manuscript.

Reviewer2: Dear Reviewer, we have incorporated all of your comments into our revision, many thanks for your help in improving our research article.

---

## [Decision Letter · Decision Letter 1]

25 Oct 2023

PONE-D-23-30535R1A Lightweight and Robust Authentication Scheme for the Healthcare System using Public Cloud ServerPLOS ONE

Dear Dr. Abbasi,

Thank you for submitting your manuscript to PLOS ONE. After careful consideration, we feel that it has merit but does not fully meet PLOS ONE’s publication criteria as it currently stands. Therefore, we invite you to submit a revised version of the manuscript that addresses the points raised during the review process.

We look forward to receiving your revised manuscript.

Kind regards,

Vincent Omollo Nyangaresi, Ph.D

Academic Editor

PLOS ONE

Journal Requirements:

Additional Editor Comments:

I have gone through the revised manuscript, and have noted that you have failed to satisfactorily address my previous comments. My previous comments, including other emergent issues include the following:

1)In the abstract, you need to describe the current solutions to public cloud server security issues. Thereafter, explain the challenges of these current solutions. This will help readers see the gap that this paper sought to fill.

2)"...The proposed protocol is verifiably secure; this article has proved it using a well-known technique, i.e.,

BAN logic...."

You may state the above as follows:

"The BAN logic and ProVerif security proofs carried out demonstrate that the proposed protocol is verifiably secure"

3) Avoid the usage of 'etc', 'i.e'. You have to write in full.

Ensure that the conclusion section is one continuous paragraph

4)"...The proposed mechanism regarding storage overhead, communication, and computation cost is analyzed.."

Replace with he following:

"...The proposed mechanism is analyzed in terms of storage overhead, communication and computation costs...."

5)"...Upon comparing the proposed protocol with prior work in terms of security and performance metrics, it has been demonstrated that the proposed scheme is superior to its competitors...."

Provide percentage improvements attained by your scheme in terms of storage, communication and computation costs.

6)"...A fuzzy extractor method is used to design the proposed protocol to make the authentication process more secure, and the system shows uniqueness while performing any task. Because an adversary cannot forge,

extract, or collide the hash image generated from user biometrics in combination with a random key extracted before authentication...."

Avoid starting your sentences with "Because...."

Reviewers' comments:

Reviewer's Responses to Questions

**Comments to the Author**

1. If the authors have adequately addressed your comments raised in a previous round of review and you feel that this manuscript is now acceptable for publication, you may indicate that here to bypass the “Comments to the Author” section, enter your conflict of interest statement in the “Confidential to Editor” section, and submit your "Accept" recommendation.

Reviewer #1: All comments have been addressed

Reviewer #2: All comments have been addressed

2. Is the manuscript technically sound, and do the data support the conclusions?

Reviewer #1: Yes

Reviewer #2: Yes

3. Has the statistical analysis been performed appropriately and rigorously? 

Reviewer #1: Yes

Reviewer #2: Yes

4. Have the authors made all data underlying the findings in their manuscript fully available?

Reviewer #1: Yes

Reviewer #2: Yes

5. Is the manuscript presented in an intelligible fashion and written in standard English?

Reviewer #1: Yes

Reviewer #2: Yes

6. Review Comments to the Author

Reviewer #1: (No Response)

Reviewer #2: All comments have been addressed and the authors answer all comments. I recommend accepting the article

7. PLOS authors have the option to publish the peer review history of their article (what does this mean?). If published, this will include your full peer review and any attached files.

Reviewer #1: No

Reviewer #2: **Yes: **Zaid Ameen Abduljabbar

---

## [Author Response · Author response to Decision Letter 1]

28 Oct 2023

Dear Editor, and Reviewers many thanks for your valuable comments provided as minor revision which improved our manuscript. We have incorporated all of your comments into our revised manuscript.

---

## [Editor Report · Decision Letter 2]

2 Nov 2023

A Lightweight and Robust Authentication Scheme for the Healthcare System using Public Cloud Server

PONE-D-23-30535R2

Dear Dr. Abbasi,

We’re pleased to inform you that your manuscript has been judged scientifically suitable for publication and will be formally accepted for publication once it meets all outstanding technical requirements.

Kind regards,

Vincent Omollo Nyangaresi, Ph.D

Academic Editor

PLOS ONE
---

## [Editor Report · Acceptance letter]

28 Nov 2023

PONE-D-23-30535R2 

A Lightweight and Robust Authentication Scheme for the Healthcare System using Public Cloud Server 

Dear Dr. Abbasi:

I'm pleased to inform you that your manuscript has been deemed suitable for publication in PLOS ONE. Congratulations! Your manuscript is now with our production department. 

Kind regards, 

on behalf of

Dr. Vincent Omollo Nyangaresi 

Academic Editor

PLOS ONE